# Fused Filament Fabrication of Slow-Crystallizing Polyaryletherketones: Crystallinity and Mechanical Properties Linked to Processing and Post-Treatment Parameters

**DOI:** 10.3390/polym16233354

**Published:** 2024-11-29

**Authors:** Lucía Doyle, Xabier Pérez-Ferrero, Javier García-Molleja, Ricardo Losada, Pablo Romero-Rodríguez, Juan P. Fernández-Blázquez

**Affiliations:** 1IMDEA Materials Institute, C/Eric Kandel, 2, Getafe, 28906 Madrid, Spain; lucia.doyle@imdea.org (L.D.); javier.garcia@imdea.org (J.G.-M.); 2AIMEN Technology Centre, O Porriño, 36418 Pontevedra, Spain; xabier.perez@aimen.es (X.P.-F.); ricardo.losada@aimen.es (R.L.); pablo.rodriguez@aimen.es (P.R.-R.)

**Keywords:** Polyaryletherketones, fused filament fabrication, 3D printing, annealing, interlayer adhesion, porosity, mechanical properties

## Abstract

Recent advancements in thermoplastics within the polyaryletherketone (PAEK) family have enhanced additive manufacturing (AM) potential in fields like aerospace and defense. Polyetheretherketone (PEEK), the best-studied PAEK, faces limitations in AM due to its fast crystallization, which causes poor inter-filament bonding and warping. This study investigated alternative, slow-crystallizing PAEK polymers: polyetherketoneketone (PEKK-A) and AM-200, a PEEK-based copolymer. Both can be printed in an amorphous state and then annealed to improve crystallinity and mechanical properties. Despite their potential, these materials have been minimally explored for AM. Our analysis compared the mechanical performance of as-printed and annealed samples and showed that slow-crystallizing PAEKs outperform fast-crystallizing PEEK. As-printed PEKK-A and AM-200 parts reached tensile strengths of 69 MPa and 47 MPa, respectively, which are about 80% of the values for injection-molded parts. In contrast, PEEK achieves only 25% due to poor inter-layer bonding. Annealing increased crystallinity (15.7% for PEKK-A, 19% for AM-200), simultaneously leading to a coalescence of smaller pores into larger ones, which affected mechanical integrity. Annealing strengthened the printed filament direction, while Z-direction strength remained limited by interlayer adhesion. Our work provides new insights into optimizing these relationships to expand the applicability of PAEK in additive manufacturing.

## 1. Introduction

Recent developments in additive manufacturing (AM) have enhanced the technology as capable of producing polymer end-use parts with complex customized structures and functionalities, enhancing their applications in various fields. Compared to traditional manufacturing processes requiring machining, molds, and tooling, AM techniques are economically favorable and offer higher flexibility. Polyaryletherketones (PAEKs) and their composite materials have become one of the research hotspots in the AM field [1]. These high-performance thermoplastic polymers exhibit superior thermal resistance, mechanical performance, low linear expansion coefficient, and chemical stability thanks to their linear aromatic structure [2]. However, these high-performance thermoplastics require much higher melting temperatures and are challenging to process.

Fused filament fabrication (FFF) is one of the most widely used AM techniques to manufacture thermoplastics or their composites; compared to other 3D printing techniques, FFF systems require smaller setups and lower running costs [3]. FFF appeared in 1992 [4] and involved manufacturing 3D polymer parts by extruding solid filament through a computer-controlled hot nozzle onto a building plate; the toolpath of the nozzle is specifically designed to create parts fabricated layer by layer. After deposition, the extruded filament solidifies, and additional layers can be subsequently deposited. In FFF, the final mechanical properties are governed mainly by the adhesion quality at the filament scale. Adhesion is described as the succession of several steps, intimate contact, coalescence, and healing, with a random distribution of the molecular polymer chains on either side of the interface [5]. In semicrystalline polymers, coalescence occurs above the melting temperature (T_m_) of the polymer when the crystalline phases have turned amorphous, and it is strongly affected by the kinetics of crystallization and the thermal gradient during the processing. If the crystallization process is fast, the filaments solidify rapidly, and there is no time for macromolecular interdiffusion between adjacent layers. Further, imbalanced crystallization at the interface, in bulk, and in previously deposited layers leads to warpage, residual stress accumulation, and interlayer delamination [6], leading to poor quality of the printed parts.

The most widely known and explored polymer within the PAEK family is PEEK [3,6,7,8,9,10,11]. Despite its outstanding properties, the fast crystallization of PEEK [12,13] makes it very difficult to control during printing [10,11], restricting its use in AM. Amorphous or slow-crystallizing PAEKs could circumvent this issue, as their slower crystallization allows for extended macromolecular diffusion and enables printing in an amorphous state. An annealing post-treatment would produce crystallization [14], improving the mechanical properties. However, the processing and post-processing behavior of slow-crystallizing PAEKs remains largely unexplored in the literature. To address this gap, the present study investigated the impact of FFF processing and annealing on the mechanical properties of slow-crystallizing PAEKs, specifically PEKK-A and AM-200, evaluating their potential as more AM-compatible alternatives to fast-crystallizing PAEKs like PEEK. This comparative analysis, which had not been previously undertaken, evaluated the tensile properties of these polymers in both horizontal (X) and vertical (Z) build directions. Through microstructural analysis and assessment of annealing effects on crystallinity, we aimed to uncover the underlying factors contributing to the mechanical behavior of each material. Notably, the tensile strength of Z-printed samples for both PEKK-A and AM-200 approaches that of injection-molded parts, which is significantly higher than reported values for fast-crystallizing PEEK, underscoring the advantages of slow-crystallizing PAEKs in AM applications. Additionally, this study revealed the critical role of porosity in FFF-printed parts: while annealing enhances crystallinity and bulk strength, it also induces pore coalescence, leading to larger pores that weaken Z-direction strength, which remains limited by interlayer bonding. These findings provide new insights into the trade-offs in crystallization and porosity control for optimizing PAEK polymers in additive manufacturing.

## 2. Materials and Methods

### 2.1. Materials

Two different PAEK filaments were studied: PEKK 60:40 filament acquired from 3DXTECH, commercially named PEKK-A, and an LM-PAEK filament supplied by VICTREX and commercially identified as AM 200. The thermal properties according to the respective data sheet are T_g_ = 162 °C and T_m_ = 305 °C for PEKK-A and T_g_ = 151 °C and T_m_ = 303 °C for AM 200. To ensure that moisture content did not interfere with the printing process, an oven was used to dry the filaments at 120 °C for 6 h before the printing process.

### 2.2. FFF Printing

The filaments were used to manufacture specimens for tensile tests according to the ISO 527-2 standard using a commercially available printer, A2V4 by 3ntr (Oleggio, Italy). A scheme of specimen processing is shown in Figure 1. The samples were printed using a 0.5 mm nozzle with a layer height of 0.25 mm using a perimeter printing speed of 25 mm/min and an infill printing speed of 30 mm/min. Preliminary tests were undertaken to find the print parameters, ensuring that both polymers were printed in the amorphous state. The resulting print parameters used for the reported results are comprehensively compiled in the Appendix A. The X specimens (horizontal build) were fabricated lying on the print bed in the XY plane, while Z specimens (vertical build) were fabricated standing vertically on the print bed as individual samples unsupported in the XY plane.

### 2.3. Post-Treatment (Annealing)

Heat treatments of the specimens were carried out in a furnace after printing. The thermal cycle was the same for both materials. The temperature was increased from RT to 160 °C at 10 °C/min, held at 160 °C for 30 min, then raised at 10 °C/min to 200 °C, and maintained there for 90 min. Finally, the samples were cooled down at 1 °C/min until they reached ambient conditions. The annealing treatment was a two-step process, in which a low-temperature step above T_g_ was first used to encourage chain diffusion, followed by a high-temperature annealing step to induce crystallization.

### 2.4. Mechanical Testing

Tensile tests were performed to characterize the samples using a 5 kN capacity standard tensile/compression machine from Shimazdu, Kyoto, Japan. Tensile properties were evaluated according to ISO 527-2; moduli were measured at a constant speed of 1 mm/min, while tensile strengths and elongation at break were measured at 5 mm/min. At least five repeated specimens were tested, and the data were averaged.

### 2.5. Differential Scanning Calorimetry (DSC)

Differential scanning calorimetry (DSC) analyzed the polymers’ phase structure. The experiments were carried out in a Q200 DSC (TA Instruments, New Castle, DE, USA). The DSC tests were performed with N_2_ as the purge gas. The samples were cut from the fabricated specimens. Samples with a mass of 10–15 mg were put on an alumina crucible and subjected to a heating cycle from RT to 360 °C at a rate of 10 °C/min. After the first heating cycle, the temperature was held at 360 °C for 15 min, which is about 50 °C above the melting temperature, to erase the thermal history; then samples were cooled to RT at 10 °C/min, and finally, subjected to a second heating from RT to 360 °C at 10 °C/min. Crystallinity was determined by Equation (1).
(1)$$\chi =\frac{{∆H}_{endo}-{∆H}_{exo}}{130}\times 100\;\%$$
where ∆*H_endo_* and ∆*H_exo_* are the integral areas of the endothermic peak and exothermic peak, respectively, and 130 J/g is the melting enthalpy of a 100% crystalline PEEK sample [15].

### 2.6. 2D X-Ray Diffraction (WAXS)

An in-house X-ray source performed the 2D X-ray diffraction patterns (SAXSpoint 5.0, AntonPaar, Graz, Austria, λ = 1.54 Å, and Dectris Eiger2 R 1M detector). The experiments were done by transmission on the grip part of the dog bone samples for tensile tests with a diameter beam size of 3 mm. The 2D X-ray diffractograms were radially integrated to obtain the 1D diffractograms using SAXS Analysis software (Version 4.30.0.148 Release) provided by Anton Paar. Crystallinity was calculated from these 1D diffractograms by comparing the crystalline diffraction peaks and the total diffractogram area. Crystalline diffraction peaks are obtained by the subtraction of the amorphous region (using the printed samples as a reference because they are totally amorphous) to the pristine 1D diffractograms. These calculations are shown in the Appendix A.

### 2.7. X-Ray Computed Micro-Tomography

X-ray computed micro-tomography was used to examine the microstructure and quantify the porosity content (General Electric Phoenix Nanotom 160 NF, Wunstorf, Germany). The target was molybdenum, 0-mode nanofocus, and no additional filter materials were used. The device has a Hamamatsu detector of 2300 × 2300 pixels, and the selected voxel size in all measurements was 5.5 µm^3^. The voltage of the X-ray tube for the scan was 50 kV, and the current was 200 µA. Each scan was composed of 2000 projections; each one is the average of 8 radiographs taken with an exposition of 500 ms. Reconstructed volumes were treated and segmented with ImageJ (1.53k version) [16]. Furthermore, porosity quantification and volume rendering were performed with Avizo 2023.2 software.

## 3. Results and Discussion

A summary of the experimental results is compiled in Table 1. We first analyzed the effect of the processing parameters and post-treatments on the mechanical properties. Further microstructural analysis will allow for a deeper understanding of the observed behavior.

### 3.1. Effects of Print Orientation and Annealing Post-Treatment on the Mechanical Behavior

The obtained modulus and yield strength for the as-printed samples of both polymers on the X print direction reached the datasheet values, confirming that the selected process parameters (see Appendix A) were adequate. As expected, the mechanical properties dropped for samples printed in the Z direction. A characteristic feature of the FFF printed parts is anisotropy arising from the orientation of the deposited filament. When testing samples printed in the Z direction, the mechanical behavior is governed by the interlayer bonding, while for samples printed in the X direction, the bulk properties of the filament play a major role. There are a couple of major factors affecting the bonding between layers. Typically, the lower the viscosity, the better the surface wetting of the freshly printed layer on a prior printed layer. A higher mobility of polymer chains (reptation) during printing also leads to better chain diffusion and entanglement across the interface between the two layers. The chain diffusion stops when the temperature at the interface drops below T_g_, or it can be hindered by crystallization. Therefore, the time scale of the polymer chain mobility is critical in determining the interlayer bond strength. Higher polymer chain mobility across the interface leads to stronger interlayer bonding [11].

The advantages of using slow-crystallizing polymers such as PEKK-A and AM 200 versus fast-crystallizing polymers like PEEK are immediately proved by analyzing the mechanical properties in the Z direction. The resulting average tensile strength of the as-printed Z samples of PEKK-A and AM 200 reached values of 69 MPa and 47 MPa, which accounts for around 80% of the values corresponding to injection-molded samples based on the product datasheet. In contrast, for fast-crystallizing polymers like PEEK, these values are typically around 25% [17]. According to the literature, the very fast crystallization rate of PEEK leads to the quick formation of crystals on the surface of PEEK during cooling, which limits the chain interdiffusion and causes weak interlayer bonding [18].

Concerning the annealing post-treatment, for PEKK, the annealing-induced crystallinity (25% by DSC) significantly increases the strength and stiffness of the specimens printed in the X direction. For the samples printed in the Z direction, the annealing-induced crystallinity translates into significantly increasing stiffness but not strength. This can be explained as for samples printed in the X direction; the load is applied axially to the filament strands. Thus, the polymer properties govern the results. The crystallinity increases the strength and stiffness of the PEKK, which is translated into increased strength and stiffness of the X-printed dog bones. However, for the samples printed in the Z direction, the load is applied perpendicular to the filament strands, and the interlayer bonding strength governs the failure of the samples and not the polymer strength. Thus, annealing increases the E modulus, as expected, but does not translate into increased tensile strength of the part with this print direction.

In the case of AM 200, the annealing-induced crystallinity (27% by DSC) increases the average strength and stiffness of the part, but the statistical significance of this increase is challenged by the larger variability of the sample’s behavior reflected in the standard deviation. Understanding the origin of this increased variability requires a detailed assessment of the microstructure, described in Section 3.2 and Section 3.3. In summary, the origin of this variability can be explained by the increase in pore size after annealing, as reported in Section 3.3. The large pores govern the failure of the parts, which is not offset by the increase in strength and stiffness of the bulk polymer caused by annealing.

The elongation at break is below 20%, as expected for polymers whose glass transition is higher than RT, as in the case of PEKK-A and AM 200. But there are two samples in which this elongation at break was higher than 100%, both as-printed PEKK-A and AM 200 and printed in X direction, this being the stretching direction. These samples have small pores between layers; during the strain, these small pores can favor a sequential breakage of the interlayer between strands, causing the slipping of the filament and allowing higher strains. After the annealing, the pores have a larger size. Also, the intrinsic mechanical properties of the filament are higher, which favors the expected behavior, and samples broke at low strain. An example of these two different behaviors is shown in Figure 2 for PEKK-A.

### 3.2. Effects of Print Orientation and Annealing Post-Treatment on the Polymer Phase Structure

To evaluate the phase structure and potential effects of the print orientation thereof, the as-printed PEKK-A and AM 200 samples were analyzed by DSC; the thermograms are presented in Figure 3. No differences were observed in the thermograms in the base of the printing orientation; for that reason, the orientation is not included in Figure 3.

A cold crystallization transition before melting was observed in the first heating cycle. This phenomenon is typically observed in low-crystallization polymers, which reorganize themselves when passing the glass transition temperature (T_g_). For both polymers, the enthalpy of the exothermic process (cold crystallization) was similar to the endothermic process (melting). Thus, both samples presented a completely amorphous state at room temperature. For PEKK-A, cold crystallization occurred at a high temperature and in a broad exothermic peak, which overlapped with the subsequent melting process. For AM 200, cold crystallization occurred at a lower temperature in a narrow exothermic peak, indicating that the crystallization of AM 200 is favored compared to that of PEKK-A.

After annealing, the first heating cycles of PEKK-A and AM 200 presented significant changes with respect to as-printed samples (Figure 3, right). In both cases, the thermograms did not show cold crystallization, and then both samples reached the maximum crystallinity possible, 24% for PEKK-A and 27% for AM 200 specimens (cf. Table 2).

Our annealing process had two steps, the first at 160 °C (30 min) and the second at 200 °C (90 min). These two steps produced two different kinds of crystals: at low temperature, smaller crystals melt at lower temperatures and the small endothermic peaks centered around 220 °C [19]; at 200 °C the crystallization is faster and larger crystals are produced, which melt at 301.1 °C for AM 200 and 290 °C for PEEK-A. In this second polymer, a third endothermic peak at 307.3 °C appeared; this one is due to the melting–recrystallization–melting process. If we observed the cold crystallization in the first heating for PEKK-A, whose onset is at 230 °C, then both isothermal crystallization steps were at lower temperatures. Then, even isothermal crystallization at 200 °C was slow and produced small crystals that melted at 290 °C, the second endothermic peak of the PEKK-A melting curve. The polymer can recrystallize at that temperature, forming larger crystals that melt at 300 °C, corresponding with the third endothermic peak at the highest temperature.

Table 2 displays the measured and calculated parameters for the first heating cycle of as-printed and annealed samples.

A deeper analysis of crystallinity and crystal orientation was performed by 2D X-ray diffraction. Figure 4 shows the 2D X-ray for both polymers, before and after the annealing and in both printing directions; colors represent the intensity of the diffraction following a rainbow scale from blue (low intensity) to red (high intensity). Only a broad circular ring with constant intensity was observed in all of the as-printed samples. However, after the annealing, four narrow circular rings appeared for PEKK-A and three for AM 200 over this broad circular diffraction due to the isothermal crystallization. No relevant differences were observed between samples with different printing directions. Also, no variations were observed in the intensity along the perimeter in any diffractions, such as for PEKK-A and AM200. This fact indicates the absence of any preferential crystallization direction, suggesting no preferred orientation of polymer chains during the printing process. Hence, the filament exhibited isotropy post-printing, implying that the mechanical properties of the printed filaments will be consistent in both the printed and perpendicular directions.

Figure 5 shows the radial integration for all 2D patterns shown in Figure 4. The left graph shows the four 1D diffractograms for PEKK-A, and the right graph shows their equivalent for AM 200. For as-printed samples, just a broad peak is observed due to the amorphous halo; there were no sharp diffractions, so the four printed samples are totally amorphous, consistent with the findings of the DSC. After annealing, the four annealed samples presented the main diffraction peaks at q = 13.1, 14.4, 15.9, and 20.1 nm^−1^, which are associated with the (110), (111), (200), and (211) planes, respectively [20]. This pattern is commonly observed for other PAEKs due to an orthorhombic unit cell [21]. In the case of PEKK-A, there was an extra diffraction peak at lower q, namely, 11.1 nm^−1^. This diffraction is associated with a second morphology (form II) but orthorhombic lattice, and it is associated with high chain stiffness and low molecular mobility [22,23]. It has been reported that this form II is preferred in PEKK-A from cooling crystallization due to the lower polymer mobility compared to isothermal crystallization from the melt at higher temperatures [24]. The crystallinities of annealed samples were calculated with the relation between the crystalline diffraction peak area and the total diffraction area, as shown in the Appendix A. The areas of crystalline diffractions were obtained after the subtraction of a normalized amorphous halo, using the as-printed samples from the annealed diffractograms. The crystallinities obtained by PEKK-A were 15.8% for the X direction and 15.7% for the Z direction. In the case of AM 200, the crystallinities were slightly higher: 19.0% for the X direction and 19.5% for the Z direction. The 2D X-ray pattern revealed that our printing conditions produced amorphous samples without polymer chain orientation. After the annealing, the crystallinity of AM 200 was higher than that obtained for PEKK-A, and the printing direction did not affect the crystallinity.

### 3.3. Effects of Print Orientation and Annealing Post-Treatment on the Porosity of the Parts

In addition to the sample crystalline morphology, the sample porosity analysis is crucial for understanding the mechanical behavior. This analysis was performed by X-ray tomography. The volume studied in all cases was the broad part of the tensile test specimens, the region useful for fixing by tensile clamps. Figure 6 shows perpendicular slices to the length of the specimens for PEKK-A and AM 200 before and after the annealing. The first relevant observation is that the pores follow the filament pattern, indicating that they were formed during printing. No pores were found inside the filaments.

The porosity for the X printing direction is higher in AM 200; however, in the Z direction, the area of the pores is higher for PEKK-A.

Another general observation is that the size of the pores increased after the annealing, independently of the polymer and the direction of the printing.

A detailed study of the porosity requires 3D analysis. Figure 7 shows the images for the pore volumes, and Table 1 shows the numerical analysis of these volumes. This confirms our previous observation that the annealed samples present larger pore sizes; this suggests that small pores could coalesce during the annealing process, forming longer ones. However, the evolution of the global pore volume depended on the material; for PEKK-A, the porosity increased with the annealing. Nevertheless, the global porosity decreased in AM 200 during the annealing.

The pore volume distribution for all analyzed samples is presented in Figure 8a,b.

In all cases, the median pore volume is <1·10^5^ µm^3^, and 75% of all the pores have a pore volume < 2 × 10^5^ µm^3^. However, there is a large dispersion in the pore size, which causes the mean to shift out of the 75% interval. This highlights the value of a volumetric analysis since it is the outlier, larger pores that will favor the failure of the part.

For both polymers, pore size is smaller in the X print direction than in the Z print direction; this could be associated with the control of the temperature because the X direction is printed on the bed, and the height of the samples is small; however, in Z print, most of the layers are at a long distance from the bed.

We can see in Figure 8a that AM 200 samples have a larger number of large pores than PEKK samples for all cases.

We can also observe that annealing increases the pore size dispersion for all cases, shifting the mean pore size to higher values.

The effect of the annealing was different for PEKK-A and AM 200; for PEKK-A, the porosity was slightly lower, and for AM 200, the porosity was doubled. This fact has to be related to the rheological behavior during the crystallization process. PEEK-A has a cold crystallization at a higher temperature. During the annealing, the samples were kept in a viscous state, in which the higher molecular chain mobility could favor the reduction of the global porosity [25]. However, the cold crystallization, densifying the sample, favored merging the pores, increasing their size and reducing the total number of pores. The faster cold-crystallization process that AM 200 suffered suggests that only densification was possible during the cold crystallization since the period of time with high polymer chain mobility was quite short; for that reason, the pore size increased, as well as the global porosity (cf. Table 1).

## 4. Conclusions

This work analyzed the effect of processing (build orientation) and post-processing (annealing) on the mechanical performance of 3D printed parts produced from commercially available filaments of the slow-crystallizing PAEKs PEKK-A and AM 200 by fused filament fabrication (FFF). Samples fabricated with these materials have been mechanically characterized by conducting tensile tests, and the observed differences are explained through exhaustive microstructural characterization of the samples. From the analysis of the results, the following can be concluded:Despite being printed in the amorphous phase, parts printed in the X direction still have superior mechanical properties over parts printed in the Z direction for both polymers. A smaller pore size is observed in this print direction, which may arise from the lower temperature gradient experienced by dog bone samples printed in the X direction versus the Z direction. Further, since the Z-printed samples were tested with the load applied perpendicular to the filament strands, the failure of the samples was governed by the interlayer bonding strength.AM samples of both slow-crystallizing PAEKs printed in the X direction present the same tensile properties as their injection-molded counterparts, based on the data sheet values. Samples printed in the Z direction, governed by the interlayer bonding strength, maintain 80% of the tensile strength in relation to the injection-molded valued. This is significantly higher than previously reported for slow-crystallizing PEEK, confirming the benefit of slow-crystallizing PAEKs for additive manufacturing arising from their amorphous state while printing.Pores appear parallel to the filament strands. They are related to poor bonding, reducing the mechanical performance in both printed directions, more important in the Z direction but also observed in the X direction. This porous interlayer in the X direction can favor higher elongation at break since the filament can slide between them during a sequential break of their interfaces. Processing adjustments are required to reduce porosity, particularly for AM 200. These could include increases of nozzle, chamber, and bed temperature; a reduction of infill; an increase of infill overlap; and a reduction of layer height. Additionally, external heating of the parts during printing is being explored.Annealing produced crystallization of the bulk polymers, as expected. No orientation of the chains was observed; therefore, the mechanical properties of the filaments are isotropic. At the same time, with annealing, pores coalesce, creating larger pores.While crystallinity strengthens and stiffens the bulk polymers, this only translates into improved strength and stiffness of FFF parts when the porosity is low. At a certain porosity level, the increased strength and stiffness of the bulk polymer cannot counterbalance the weakness produced by porosity. In our study, we observed that even for parts printed in the X direction when the porosity was in the range of 5% (the case of AM 200), crystallinity could not offset the weakness caused by porosity. Further research is needed to better determine the porosity threshold at which annealing can be beneficial.

The results reported here confirm the advantage of slow-crystallizing PAEKs for AM applications and further provide new insights concerning the effects that porosity and crystallinity have on the mechanical behavior of FFF parts, supporting the optimization of printing parameters, printing direction, and post-processing of amorphous and slow-crystallizing PAEK additive manufacturing polymers.

## Figures and Tables

**Figure 1 polymers-16-03354-f001:**
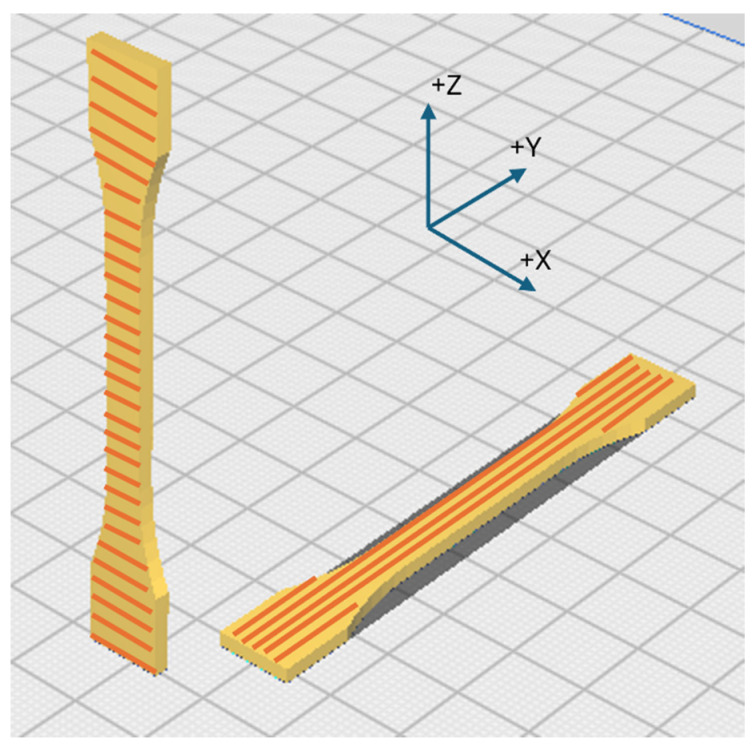
Scheme of the 3D printed process for tensile test samples; lines represent the filament pattern. XY is the plane of the 3D chamber bed.

**Figure 2 polymers-16-03354-f002:**
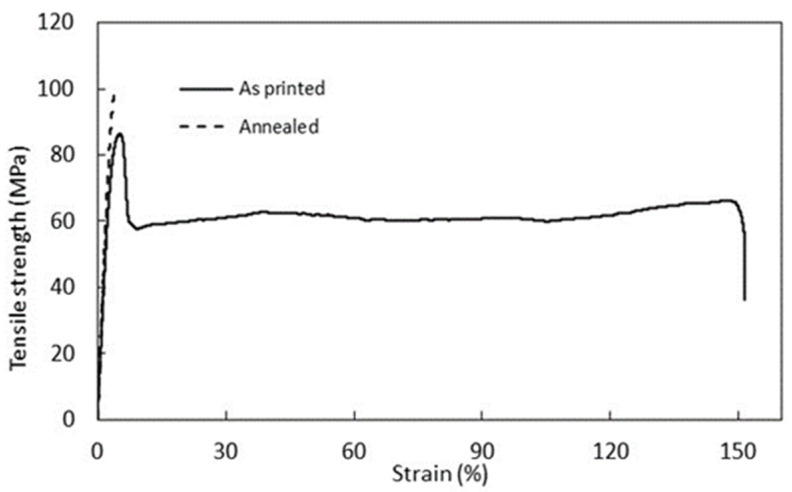
Stress-strain curve for the PEKK-A (X) printed samples.

**Figure 3 polymers-16-03354-f003:**
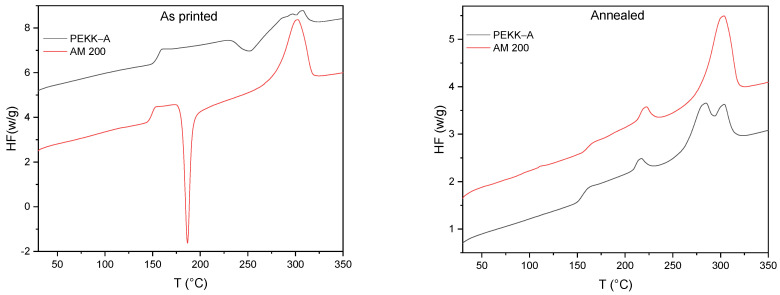
DSC, first heating cycle for as-printed” samples (**left**) and annealed samples (**right**). Exo down.

**Figure 4 polymers-16-03354-f004:**
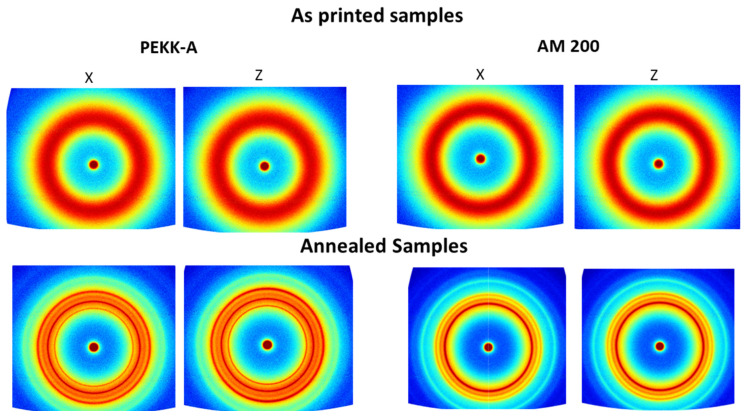
2D X-ray pattern for PEEK-A and AM 200, as printed and after annealing, and in both printing directions. Blue color → low intensity (a.u.); red color → high intensity (a.u.).

**Figure 5 polymers-16-03354-f005:**
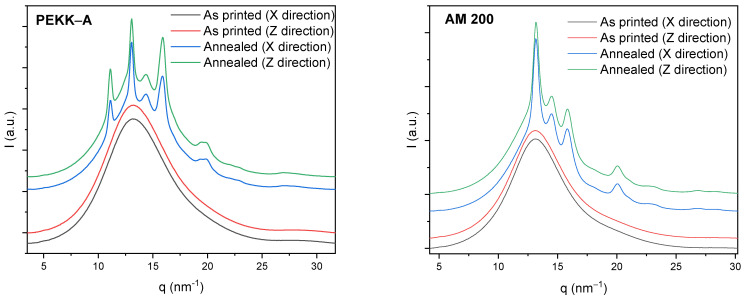
Radial integration for all 2D X-ray patterns. (**left**) The diffractograms for PEEK-A; (**right**) the diffractograms for AM 200, for samples “as printed” and after annealing and in both printing directions. X and Y indicate the printing directions.

**Figure 6 polymers-16-03354-f006:**
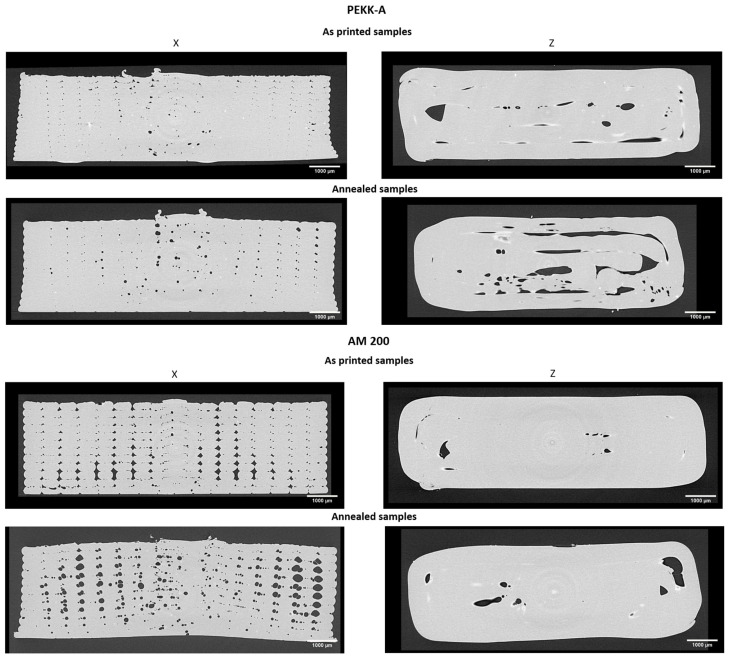
X-ray tomography perpendicular slices to the length of tensile specimens for PEKK-A samples (**upper**) and AM 200 (**lower**) before and after the annealing. X and Z indicate the printing directions.

**Figure 7 polymers-16-03354-f007:**
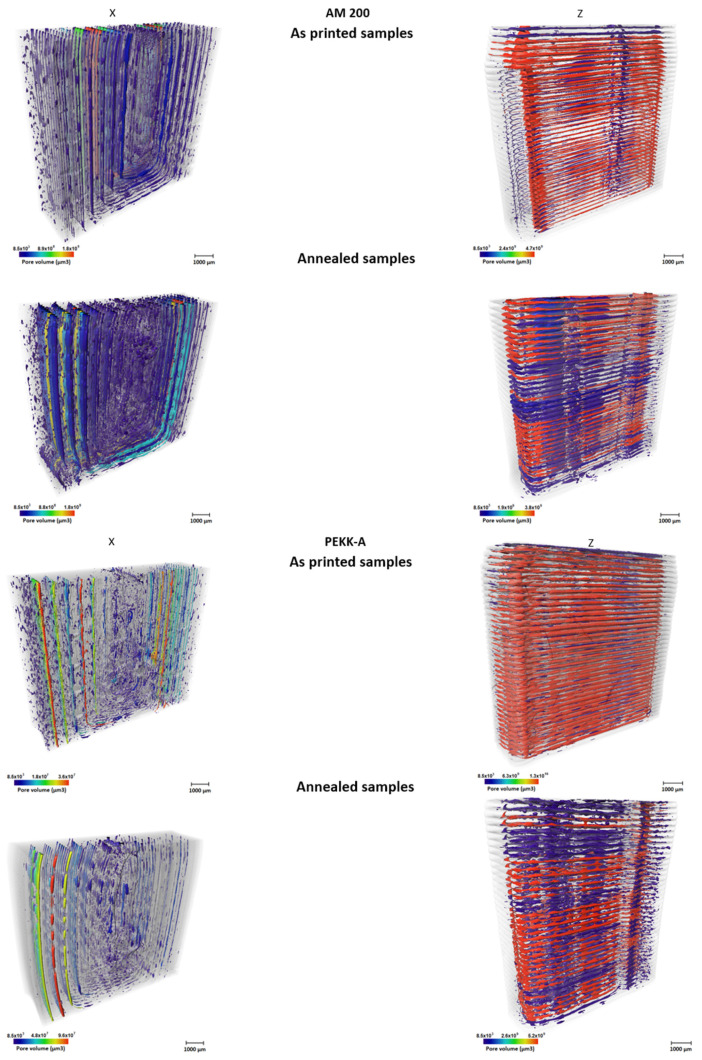
Renders of the pores volume distribution taken by X-ray tomography on tensile specimens for PEKK-A samples (**left**) and AM 200 (**right**) before and after the annealing. X and Z indicate the printing directions. Pores are colorized following their volume: the biggest pores are red, and the smallest pores are dark blue. The rest of the material is set to semitransparent gray values.

**Figure 8 polymers-16-03354-f008:**
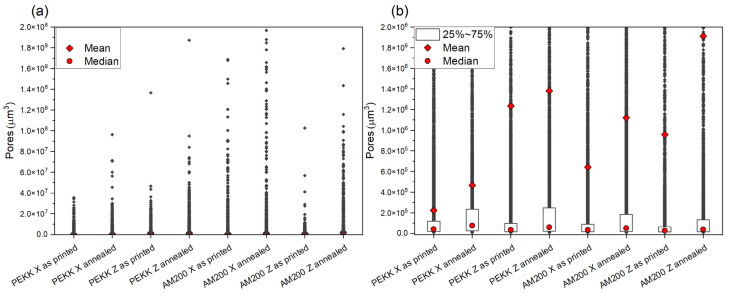
Pores contained in each sample. (**a**) All the measured pores, in volume, to capture the extent of the large pores. (**b**) Close-up to visualize the mean and median values and the size, including the 25–75% interval.

**Table 1 polymers-16-03354-t001:** Compiled Results.

Polymer	Processing	Printing Direction	Tensile Strength (MPa)	E Modulus (GPa)	Crystallinity (%)	Pore Volume Fraction (%)	Max. Pore Volume (µm^3^)	Mean Pore Volume (µm^3^)	Median Pore Volume (µm^3^)
PEKK-A	As printed	X	84 ± 3	3.2 ± 0.3	0	1.5 ± 0.3	3.56·10^7^	2.2·10^5^	6·10^4^
		Z	69 ± 2	2.5 ± 0.2		6 ± 3	1.26·10^10^	1.2·10^6^	3.5·10^4^
	Annealed	X	96.4 ± 10.0	3.6 ± 0.1	15.75	1.2 ± 0.3	9.64·10^7^	4.6·10^5^	7.7·10^4^
		Z	64 ± 5	2.9 ± 0.2		5 ± 2	5.17·10^9^	1.4·10^6^	6·10^4^
AM-200	As printed	X	60.7 ± 0.8	2.7 ± 0.7	0	5 ± 2	1.77·10^9^	6.4·10^5^	3.3·10^4^
		Z	47±	2.2 ± 0.2		2 ± 2	4.74·10^9^	9.6·10^5^	5.1·10^4^
	Annealed	X	71 ± 8	2.9 ± 0.5	19	9 ± 2	1.76·10^9^	1.1·10^6^	5.1·10^4^
		Z	49.3 ± 10.0	2.5 ± 0.5		5 ± 3	3.80·10^9^	1.9·10^6^	3.8·10^4^

**Table 2 polymers-16-03354-t002:** Transitions temperatures, enthalpies, and estimated crystallinities correspondent to the first heating cycle. *: In the case of as-printed samples, this enthalpy is (${∆H}_{m}-{∆H}_{cc})$.

	Material	T_g_ (°C)	T_cc_ (°C)	T_m_ (°C) (For Each Peak If Multiple)	∆*H_m_* (J/g) *	Crystallinity (%)
As printed	PEKK-A	157	252.7	307.3	0	0
	AM 200	150	186.7	301.1	0	0
Annealed	PEKK-A	156	-	216.7/290/303.8	29.9	24.7
	AM 200	159	-	222.0/302.9	35.6	27.4

## Data Availability

The original contributions presented in this study are included in the article/Appendix A. Further inquiries can be directed to the corresponding author.

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
