# Peer review of "Fused Filament Fabrication of Slow-Crystallizing Polyaryletherketones: Crystallinity and Mechanical Properties Linked to Processing and Post-Treatment Parameters"

_polymers, 2024, doi:10.3390/polym16233354_

Round 1
Reviewer 1 Report
Comments and Suggestions for Authors
Doyle et al., have presented the research article titled “Fused Filament Fabrication of amorphous and slow-crystalizing PAEKs. Crystallinity and Mechanical Properties Linked to Processing and Post-Treatment Parameters”. Herein this article authors have provided the comprehensive comparison of two different Polyarylether ketones polymers (PAEKs): PEKK 60:40 and 19 LM-PAEK (AM 200) fabricated by fused filament fabrication technique with mechanical properties. Overall the details are very well described, article is acceptable but there are few suggestions which should be addressed before being considered for publication.
1. Abstract section has no attraction, mostly general discussion, there should be the little description of the mechanical properties outcomes (values), tensile strength, Transitions temperatures, enthalpies, and estimated crystallinities. These mentioned outcome values will provide the authors the better understanding to cite this work.
2. In the introduction section, there is much discussion about the Polyarylether ketones polymers and its different types. But there is no literature survey related to presented study, how the readers and researcher compare the presented work with the literature.
3. In the introduction section, clarity of objectives should be well stated, providing a more explicit overview of the specific research questions or hypotheses which can enhance the reader's understanding of the study's focus.
4. It is important to clearly emphasize the novelty of the study compared to existing research. Highlight what distinguishes the present research from the literature.
5. In the materials section 2.2 FFF Printing, the parameters used by the authors are taken from the literature?
6. Throughout the manuscript, there remain the question in my mind that authors have not empathized at all on the practical approach. I suggest the authors that conclusion should benefit from a more robust summary of key findings and their practical implications.
Reviewer 2 Report
Comments and Suggestions for Authors
1. What does it mean in the abstract “The most widely explored AM PAEK is PEEK.”?
2. It is not clear if you are investigating PAEK, PEKK, or PEEK. 3D printing of PEKK would be less challenging than PEEK.
3. What are the challenges in printing these types of materials and what were your strategies to improve their printabilities?
4. Do not use abbreviations in the abstract without introducing them in advance.
5. The abstract needs to be rewritten. It seems like an AI tool has written the abstract.
6. The novelty of the work shall be highlighted in the last paragraph of the introduction section.
7. What is the merit, and motivation for studying the specific materials?
8. Page 2, “The printing settings were adjusted for both materials to be printed in an amorphous state.” How did you control the printing parameters and monitor the amorphous state?
9. Why did you print “Z specimens (vertical build)” in the direction shown in Figure 1? If you wanted to measure the tensile strength in horizontal and perpendicular directions of printing patterns, why did you not adjust the printing patterns perpendicular to the dog-bone specimens?
10. For each test conducted in this work, you need to justify why you did it.
11. How did you calculate the “Crystallinity (%)” in Table 1? Why was it ZERO for as printed specimens?
12. Printed PEKK, PEEK, and PAEK specimens show an almost brittle failure, while Figure 2 shows a strain of about 150%.
13. What is shown in Figure 4? What are those colors?
14. According to Figure 4, The annealed specimens have higher voids, while they showed higher tensile strength. Explain why?
15. You need to discuss the results not just report them.
16. There are many works studying the crystallinity of PEKK,PEEK, and PAEK materials such as the below ones. It would be best if you discussed them in the introduction section.
Advancements in functionally graded polyether ether ketone components: Design, manufacturing, and characterisation using a modified 3D printer
Comparison of various 3D printed and milled PAEK materials: Effect of printing direction and artificial aging on Martens parameters
Round 2
Reviewer 2 Report
Comments and Suggestions for Authors
The paper is accepted in its current form.